# The biased hand. Mouse-tracking metrics to examine the conflict processing in a race-implicit association test

**Michael Di Palma**[1]*, **Desiré Carioti**[2], **Elisa Arcangeli**[2], **Cristina Rosazza**[2],
**Patrizia Ambrogini**[3], **Riccardo Cuppini**[3], **Andrea Minelli**[3], **Manuela Berlingeri**[2,4,5]*

1 Department of Experimental and Clinical Medicine, Università Politecnica delle Marche, Ancona, Italy,
2 Department of Humanities Studies, University of Urbino Carlo Bo, Urbino, Italy, 3 Department of
Biomolecular Sciences, University of Urbino Carlo Bo, Urbino, Italy, 4 NeuroMi, Milan Centre for
Neuroscience, Milan, Italy, 5 Center of Developmental Neuropsychology, ASUR Marche, Pesaro, Italy

* m.dipalma@staff.univpm.it (MDP); manuela.berlingeri@uniurb.it (MB)

## Abstract

In this study, we adapted a race-Implicit Association Test (race-IAT) to mouse-tracking (MT) technique to identify the more representative target observed MT-metrics and explore the temporal unfolding of the cognitive conflict emerging during the categorisation task. Participants of Western European descent performed a standard keyboard-response race-IAT (RT-race-IAT) and an MT-race-IAT with the same structure. From a behavioural point of view, our sample showed a typical Congruency Effect, thus a pro-White implicit bias, in the RT-race-IAT. In addition, in the MT-race-IAT, the MT-metrics showed a similar Congruency Effect mirroring the higher attraction of the averaged-trajectories towards the incorrect response button in incongruent than congruent trials. Moreover, these MT-metrics were positively associated with RT-race-IAT scores, strengthening the MT approach's validity in characterising the implicit bias. Furthermore, the distributional analyses showed that mouse trajectories displayed a smooth profile both in congruent and incongruent trials to indicate that the unfolding of the decision process and the raised conflict is guided by dynamical cognitive processing. This latter continuous competition process was studied using a novel phase-based approach which allowed to temporally dissect an Early, a Mid and a Late phase, each of which may differently reflect the decision conflict between automatic and controlled responses in the evolution of the mouse movement towards the target response. Our results show that the MT approach provides an accurate and finer-grained characterisation of the implicit racial attitude than classical RT-IAT. Finally, our novel phase-based approach can be an effective tool to shed light on the implicit conflict processing emerging in a categorisation task with a promising transferable value in different cognitive and neuropsychological fields.

## Introduction

Our ability to readily categorise the myriad of contents of the world represents an efficient adaptive processing tool to provide maximum information with minimum cognitive effort [1,

**Data Availability Statement:** The raw data processed and analysed for this work are available at the Open Science Framework (OSF, https://osf.io/B2JC5/). Further information regarding the

experimental design and statistical analyses are reported in the Supporting Information files.

**Funding:** The authors received no specific funding for this work.

**Competing interests:** The authors have declared that no competing interests exist.

2]. In particular, this energy-saving ability allows us to navigate our social environment efficiently by adopting fast and relatively inflexible strategic (heuristic) judgments of other people [3]. Making decisions and categorising other people by using these heuristics, based on simple associations such as sex or race, may not always result in a 'socially correct' behavioural response bringing out implicit stereotypes and racial prejudice towards the members of a social group [4]. From a theoretical point of view, an implicit stereotype may be defined as overgeneralised semantic attributes automatically associated with a particular social group, whereby within the semantic memory associative network system, a social concept (e.g. race or gender) is firmly combined with an attribute concept (e.g. negative or positive) [5]. Thus, the activation of a social concept spreads out to associated attribute concepts in the network, making them more easily accessible during judgements to all group members of that social group. Employing this theoretical framework, a stereotypical association (such as 'Black' and 'negative') stored in semantic memory might be automatically activated, eliciting an implicit stereotype effect and a consequent faster behavioural outcome [6].

Based on these assumptions, the use of indirect measures such as reaction times (RTs) has quickly called great attention to study these cognitive biases characterising the implicit domain of social cognition [7]. In this regard, one of the most influential and widely used research paradigms to study stereotypical association is the Implicit Association Test (IAT) [8]. In its original form, the IAT is a stimuli-association reaction time test, in which *concept stimuli* (e.g. pictures depicting the actors' faces of a different race) and *attribute stimuli* (such as negative or positive words) are sequentially presented on a computer screen. Participants are asked to categorise the stimuli (either pictures, words, or both) by pressing one of two keyboard keys. For *stereotype-congruent trials*, in which concept stimuli are paired with attribute stimuli having the same stereotypical implicit valence, participants are instructed to respond with one keypress to stimuli representing White race and positive words and with another keypress to stimuli representing Black race and negative words, for example. On the contrary, in *stereotype-incongruent trials*, the implicit valences of concept stimuli and attribute stimuli are opposite, and for example, participants are asked to use one keypress to categorize stimuli representing White race and negative words, while another keypress to stimuli representing Black race and positive words. The outcomes of interest are mainly the RTs to a different combination of stimuli pairings. The basic assumption of the test is that participants' performance during the categorisation task is a function of implicit association strength between the two paired categories. Therefore, if there is a significant *Congruency Effect* or *IAT Effect*, characterised by longer RTs for incongruent trials than congruent, we may conclude that White-positive/Black-negative words associations are stronger than the inverse associations [8]. Finally, as assumed by the associative network model, the association strength between concept and attribute stimuli may be considered a reflection of one's implicit attitude towards that categorisation [5]. Based on this logic and its flexible application to any stimuli pair, the IAT has been used as an implicit measure for a wide range of different social categories, such as gender and ethnicity [9]. Despite the enormous popularity of this test, the indexes generated by the standard RT-IAT procedure are not without limitations. In particular, one of the drawbacks of RTs is that they may be considered a categorisation '*black box*', namely a mere endpoint that tells us little about all the ongoing decision-making processes underlying categorisation dynamics [10].

Recent developments using continuous measures, able to follow the unfolding of decision-making processes over time, may represent a promising alternative to overcome RT-IAT limitations. Most notably, mouse-tracking (MT), an emerging real-time technique of measuring the computer-mouse movements made by participants while taking a decision, has proven to be very useful in unveiling the online decision-making dynamics underlying the social

categorisation process [11–14]. Throughout the continuous capturing of motor trajectories during the entire response process, the MT approach offers a data-rich spatiotemporal real-time window into decision-making processes [15]. In classic RT-IAT, participants are asked to conclude a standard IAT experiment in which longer RTs for incongruent than congruent trials reflect a relatively greater amount of '*decision conflict*' between the concept/attribute association for incongruent than congruent trials, as mirrored by an evident *Congruency* or *IAT Effect* [16]. However, looking at the unfolding of a hand movement towards a target position, we could also explore the dynamic evolution of the '*decision conflict*' from its emergence to its resolution. When moving a computer mouse to reach a target position on the PC screen, a continuous interplay between the prediction of the movement and the actual state of the motor system takes place [17]. By making strategic corrections of movements during the temporal unfolding of a decision conflict, these internal models of motor control, allow us to reach the goal and thus respond correctly during a categorisation task [17–20]. All these strategic motor corrections may be considered as reflecting the amount of the conflict generated during the unfolding of a categorisation task, and they may be collected and analysed using the MT approach [15, 21, 22]. Several neuropsychological studies considered hand movement as a proxy to detect and quantify the intertwined *automatic* and *controlled* motor processes needed to update actions [17–20]. Under these assumptions, in the past decade, the MT approach has become a popular method to document different aspects of conflict processing underlying decision making [12, 15, 21–26]. In this line of reasoning, we aim to extend the evidence mentioned above using the continuous nature of mouse trajectories collected during *stereotypical-congruent* and *incongruent trials* to temporally dissect the unfolding of the race-IAT *Congruency Effect* applying a novel phase-based approach. We started from the '*conflict monitoring hypothesis*' [27, 28] and from Erb's proposal [29] to design our phase-based approach. Accordingly, we individuated three phases: i) the *Early Phase*, as the time interval between stimulus appearance and the onset of mouse movements (i.e., the 'Initiation Time'; [29]), which plausibly corresponds to the *starting point of the decision conflict*; *ii*) the *Mid Phase* as the time interval between the movement onset and the trajectory point of highest attraction towards the incorrect response, mirroring the dynamical competition between *automatic* and *controlled* processes in guiding the motor strategy selection [18–20], in line with Erb's suggestions [29]; *iii*) the *Late Phase* as the time needed to reach the selected target response reflecting the predominance of *controlled* motor corrections during task execution [26].

Although MT-paradigms have been widely used to characterise the underlying dynamics of various decision conflict phenomena [12, 14, 24, 30], only two experimental studies incorporated MT-measures in an IAT structure [31, 32]. These studies have pointed out a clear-cut Congruency Effect using different temporal (e.g. RTs [31]), dynamic (e.g. Velocity [31, 32]) and geometrical (e.g. MD and AUC [31, 32]) MT metrics. In addition, the MT metrics allow detecting whether, during the temporal unfolding of mouse movements towards the response target, the trajectories follow an abrupt profile characterised by a sharp mid-flight correction mirroring the competition between two systems during the evolution of the movement (Dual-system framework; [33, 34]), or relatively smooth profile mirroring a dynamical competition of different conscious and unconscious processes that simultaneously compete in guiding the movement (Dynamical framework; [12, 35, 36]).

However, none of the studies mentioned above investigated the implicit racial stereotype, nor did they consider the three phases mentioned above. In this regard, recently, Melnikoff and colleagues [37] showed that the MT approach is an effective tool to capture conflict in explicit evaluations of racial groups. However, the authors measured the racial bias by employing an MT task profoundly different from the race-IAT paradigm [38]. Accordingly, they adopted the procedure developed by Wojnowicz and colleagues [38] involving Black/White

names and positive/negative concepts as stimuli, with the response options 'Like' or 'Dislike'. Participants were called to respond 'Like' to Black and White people's names and exhibited larger AUCs in responding 'Like' to Black names than White ones [37, 38]. The dissimilarity of the MT approach used by Melnikoff and colleagues [37] compared with an IAT structure makes the adoption of the MT approach in the classical race-IAT of primary interest. Indeed, incorporating MT-measures in a classical race-IAT paradigm gives us the chance to go beyond RTs and error rates, obtaining an in-depth characterisation of the individual implicit racial bias and, at the same time, a real-time window to follow the temporal unfolding of the processes behind the conflict emergence and resolution throughout the categorisation task [22, 25, 26, 39]. In addition, MT measures allow assessing the presence of bimodality in the response distribution of the stereotype-congruent and -incongruent trials for distinguishing between single-process and dual-process phenomena [40]. The logic is that if mouse-trajectories are a function of a dual-system framework based on stage-based cognitive processing, we should see an abrupt profile characterised by a consequent bimodal distribution of the MD values. Conversely, a smooth profile of the mouse-trajectories, as a function of dynamic competition of both possible responses across the unfolding of the movement, should be predicted by a unimodal distribution of the MD values [15, 26, 41].

In the present study, using a classical within-subject design, participants were asked to complete a standard RT-race-IAT and an MT-race-IAT with the same structure of the keyboard-response one to achieve the following experimental aims. First, we want to replicate the results of previous MT-IAT studies [31, 32] exploring a race *Congruency Effect*. Second, by a direct comparison between participants' performance in the RT-race-IAT with the MT one, we want to identify among the plethora of metrics gathered by the mouse trajectories' analysis those able to characterise the implicit racial bias as much as the RT-race-IAT score computed as described in Greenwald, Nosek [42]. Third, we want to explore whether the MT approach can give us a finer-grained characterisation of the 'decision conflict' between *automatic* vs *controlled* responses in line with the three-phases model described here.

## Methods

### Participants

Participants were recruited according to the following criteria: aged between 18 and 35 years (Mean Age = 23.60 y.o.; standard error of mean = 0.51 y.o.; for details see S1 File), right-handed, native Italian speakers and of Western European descent. Based on recent studies incorporating MT-measures in an IAT structure [31, 32], an *a priori* analysis with G*Power3 [43] was performed to estimate the sample size needed to explore with large effects size the race *Congruency Effect*. Accordingly, the recruitment of 24 subjects would have allowed obtaining a robust statistic power with a relatively conservative confidence level ($f$ = 0.45, 1-β = 0.80, α = 0.001). However, the sample size was increased to perform a Principal Component Analysis (PCA) on each condition of the MT-race-IAT (*White-Positive*, *White-Negative*, *Black-Positive*, *Black-Negative*), including all metrics collected. Indeed, according to established guidelines for factor analysis [44, 45], the subjects/variables ratio has to be at least equal to 5:1 for attenuating potential errors and optimising the generalizability of the obtained findings. Therefore, the sample size was incremented from 24 to 120 (57.5% female). Data from four (one male and three females) were excluded due to technical problems throughout the recording of mouse movements during the MT-race-IAT task. Before the study, all participants underwent an informed consent process, which was approved by the Ethics Board of the University of Urbino–Carlo Bo, and carried out in accordance with the Declaration of Helsinki [46].

## Handedness index

Before the experiment, participants' handedness was assessed using a modified version of the Edinburgh Handedness Inventory (EHI) [47]. Participants were instructed to indicate, by a four-digit rating scale, which hand they preferred to use for fifteen activities [48] including one specific item for computer mouse usage (-100 = exclusively left, -50 = preferably left, 0 = no preference, 50 = preferably right, 100 = exclusively right; Mean EHI score = 78.27; standard error of mean = 2.5; for details see S1 File).

## Stimuli and apparatus

The race-IAT structure was adapted from Greenwald, McGhee [8], with race (White or Black) and words (positive or negative) as the target categories. Concept stimuli for both race-IAT consisted of twenty computer-generated neutral and young male face identities (ten White's face; ten Black's face) using FaceGen Modeller 3.0 (www.FaceGen.com). Attribute stimuli for both race-IAT consisted of twenty Italian words (ten negative words: agony, war, pain, evil, hate, bad, disaster, grief, hatred, and poison; ten positive words: lucky, honour, gift, happy, love, joy, peace, peaceful, pure and loyalty). A colour-categorisation task involving circle shapes was used as a practice trial in the MT-race-IAT task [21]. Circle shapes stimuli for this practice trial consisted of twenty circles forms (ten black and ten white). For both race-IAT, we used as presentation software the Python-based experimenter builder OpenSesame 3.2 [osdoc.cogsci.nl; Mathot, Schreij [49]]. For the MT-race-IAT task, the OpenSesame mousetrap plug-in 3.1 [50] was used to record real-time mouse cursor movements (at 100 Hz) during the trials. Stimuli were presented on a 60-Hz 22-inch flat-panel display (Samsung S22F350FH), connected to a desktop computer (DELL Precision Tower 5810) running the Windows 10 operating system. For the RT-race-IAT, all participants used the same Logitech USB-keyboard (Logitech K120), and for the MT-race-IAT, all participants used the same Logitech Optical Mouse with USB connection (Logitech 910–004855 USB 6 Button Scrollwheel Optical Mouse) with mouse sensitivity set at medium speed and acceleration. All the experiments were conducted full screen with a resolution of 1,024 × 768 pixels.

## Procedure

After providing a general demographic survey, participants completed the RT and MT race-IAT tasks in counterbalanced order. Both race-IAT tasks consisted of 220 experimental trials. The experimental task structure was almost identical, except that before the beginning of the MT-race-IAT task, participants completed 40 practice colour-categorisation trials to familiarise themselves with the mouse task (see Table 1).

In each experimental trial of both race-IAT, participants categorised a stimulus from one of the following four categories: a picture of White man (*concept stimulus*), a picture of a Black man (*concept stimulus*), a positive word (*attribute stimulus*), or negative word (*attribute stimulus*). The target stimuli were organised into seven blocks. The critical blocks were the 3rd, 4th, 6th and 7th and consisted of 80 trials (40 per each block) for both stereotype-congruent (3rd and 4th blocks; white-positive/black-negative) and incongruent (6th and 7th blocks; white-negative/black-positive) trials (see Greenwald, Nosek [42]). All the blocks were sequentially displayed from the first to the seventh, the order of the trials within each block was randomly assigned using OpenSesame's backend settings, and the position order of the target response alternatives in the top-left versus the top-right of the screen was varied across trials using OpenSesame's advanced loop operations [21, 50]. At the start of the experiment, participants were instructed to respond as quickly and accurately as possible; all the instructions for the categorisation task were computer-administered at the beginning of each block. In the RT-

**Table 1. Block sequence of the both RT- and MT-race-IAT.**

| Block Number | RT-race-IAT | | | MT-race-IAT | | |
|---|---|---|---|---|---|---|
| | Stimuli | Response Button 1 | Response Button 2 | Stimuli | Response Button 1 | Response Button 2 |
| *Practice Block —1* | *No practice blocks* | | | 20 black-white circles | White | Black |
| *Practice Block —2* | | | | 20 black-white circles | Black | White |
| 1 | 20 concept pictures | White | Black | 20 concept pictures | White | Black |
| 2 | 20 attribute words | Positive | Negative | 20 attribute words | Positive | Negative |
| 3 | 20 concept pictures + 20 attribute words | White + Positive | Black + Negative | 20 concept pictures + 20 attribute words | White + Positive | Black + Negative |
| 4 | 20 concept pictures + 20 attribute words | White + Positive | Black + Negative | 20 concept pictures + 20 attribute words | White + Positive | Black + Negative |
| 5 | 20 concept pictures | Black | White | 20 concept pictures | Black | White |
| 6 | 20 concept pictures + 20 attribute words | Black + Positive | White + Negative | 20 concept pictures + 20 attribute words | Black + Positive | White + Negative |
| 7 | 20 concept pictures + 20 attribute words | Black + Positive | White + Negative | 20 concept pictures + 20 attribute words | Black + Positive | White + Negative |

Note. The critical blocks were the 3rd, 4th, 6th and 7th and consisted of 80 trials for both stereotype-congruent (3rd and 4th blocks; white-positive/black-negative) and incongruent (6th and 7th blocks; white-negative/black-positive) trials. For details see Greenwald, Nosek [42].

race-IAT, participants were instructed to categorise each target stimulus (appearing at the bottom centre of the screen) by pressing an appropriate keyboard-key to choose the target response alternative that appeared in the top-left and top-right corner of the screen ('E-key' for top-left response alternative; 'I-key' for top-right response-alternative). Participants' response time and error rates were recorded during the entire task (blocks 1 to 7). In the MT-race-IAT, the IAT procedure closely paralleled the RT one. To have a comparable starting position for all mouse trajectories, on each trial, participants clicked a 'start' button that appeared at the bottom centre of the screen, and the target stimulus would appear in its place. Participants categorised all the target stimuli by moving and clicking over the chosen target response alternative (top-left and top-right corners of the screen) with the computer mouse. If mouse movements were not initiated within 400 ms by the appearance of the target stimulus, a message displaying the word 'Faster' appeared to encourage a quicker trial categorisation. During the categorisation task, in addition to participants' response time and error rates, the x and y coordinates of mouse movements (sampling rate = 100 Hz) were recorded by using the OpenSesame mouse-trap plug-in 3.1 [50]. The entire course of a typical trial is displayed in Fig 1.

### RT-race-IAT D scores calculation

The behavioural data were collected during the RT-race-IAT, namely RTs and error rates (Accuracy) [42]. In particular, the RT-race-IAT scores were computed using an ad-hoc-created R routine to gather the participant's D score using the improved algorithm implemented by Greenwald, Nosek [42]. The D scores were coded in the direction of the association between positive words and White targets; therefore, the higher the score, the stronger the association between positive concepts and the White race unveiling a pro-White implicit bias.

### MT-race-IAT data preparation

All the mouse-tracking data collected were processed according to with standard practice [21, 30, 50]. The data processing was computed employing ad-hoc created R routines based on the

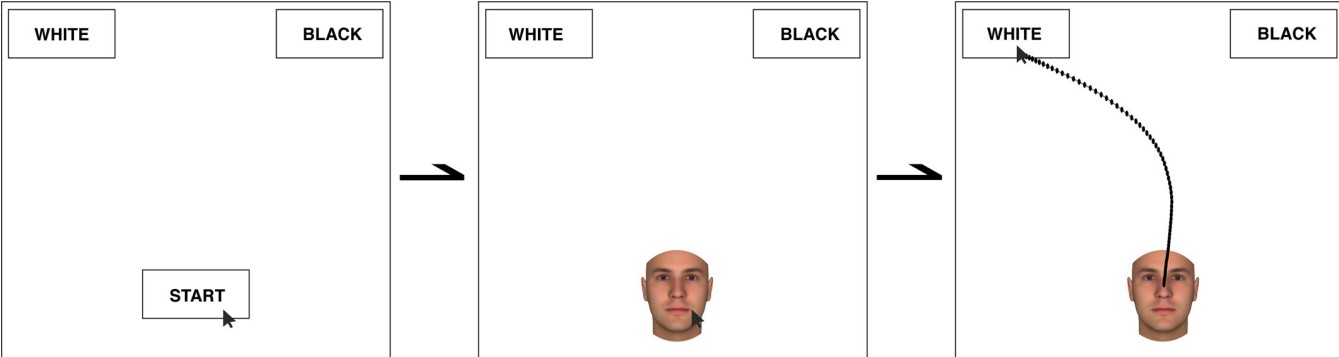

**Fig 1. Representative illustration of a typical trial of MT-race-IAT.** After clicking on 'Start' button at the bottom centre of the screen, the stimulus is presented in its position. The task consists in categorising the stimulus as fast as possible by moving and clicking over the chosen on the top-left or top-right target response alternatives with the computer mouse while the real-time mouse trajectory is recorded.

available *readbulk* [51] and *mousetrap* [50] packages. First, all the raw mouse-trajectories were time-normalised into 101 time-steps, isotropically rescaled (to preserve the *x*- and *y*-coordinates), aligned to the exact initial coordinates (namely: 0, 0) and remapped such that each trajectory terminated at the top-right response location (for a detailed description of rescaling and normalisation procedures, see Kieslich and Henninger [50]). All the data gathered by the above procedures were further processed, focusing the analysis on the trials obtained in the MT-race-IAT critical blocks, namely the 3rd and 4th as stereotype-congruent and 6th and 7th as incongruent. Specifically, incorrect trials and trajectories that were ±3 standard deviations from the mean on reaction times or an initialisation time longer than 500 ms were removed. This procedure excluded 2% of stereotype-congruent trials and 4% of incongruent trials. Then, the remaining data were used to determine temporal, geometrical, dynamic, and uncertainty MT metrics to investigate participants' performance. In particular, the temporal MT-metrics comprised the Initialisation Time (IT; this metric refers to the time in ms between the stimulus presentation and onset of mouse movement), the Reaction Time (RT; this metric is the total time in ms needed to move the mouse from the start button to the target response), the Maximum-Deviation Time (MD-Time; this metric refers to time in ms needed to reach the point of highest attraction towards the incorrect response, namely the Maximum Deviation value, from the start button) and the Time in ms at which Maximum Velocity (Vel-Max-Time) or Acceleration (Acc-Max-Time) occurred first. The geometrical MT-metrics comprised the Maximum Deviation (MD), which is the maximum perpendicular deviation of the observed trajectory from the idealised-straight path connecting the start button and target response and the Area Under the Curve (AUC), which corresponds to the geometric area recorded between the observed trajectory and idealised one. The dynamic MT-metrics comprised the *x*-maximum Velocity (Vel-Max) and Acceleration (Acc-Max). It is worth noting that the MT-metrics were calculated on time-normalised data and are thus dimensionless. Besides all the classical MT-metrics, we determined two other uncertainty values: the number of directional changes along the *x*-axis (Xpos-flips) and the sample Entropy (Entr). For a detailed description of computational procedures to determine MT-metrics, see Kieslich and Henninger [50].

### Conflict processing's temporal phases

Based on the Congruency Effect assumption that, on average, all the temporal MT metrics collected in stereotype-incongruent trials should be longer than in stereotype-congruent ones, the

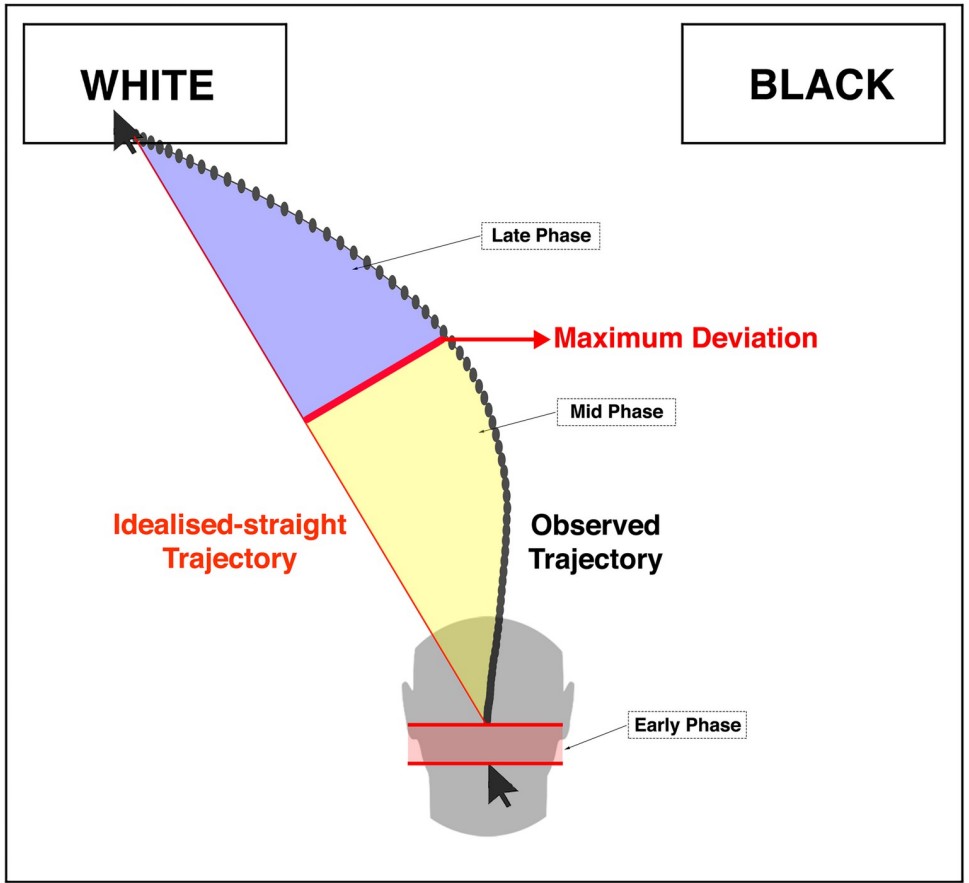

**Fig 2. Representative illustration of the temporal unfolding of a typical trial of MT-race-IAT.** The Early Phase (the latency from stimuli presentation and the onset of mouse movements) would mainly correspond to the activation of the automatic responses elicited by stimulus processing. According to Erb's proposal [29], the decision conflict between automatic and controlled responses would start here, while reaching its apex at the point of highest attraction towards the incorrect response (namely the Maximum Deviation in an MT trial). The latency needed to achieve that highest attraction from the beginning of mouse movement corresponded to the mid-phase. Following the conflictual apex, progressively controlled motor corrections would allow reaching the target. The latency needed to achieve the target response from the conflictual apex was used to obtain the Late Phase.

following temporal MT-metrics were used to explore our three phases model (see Fig 2): IT, MD-Time and RT. Specifically, for each participant, the Early Phase index was computed according to the following formula:

$$Early\ Phase\ index = IT\ incongruent\_trials - IT\ congruent\_trials$$

Then, for the Mid Phase index, we first calculated the difference between the averaged MD-time and the averaged IT values for each participant in both stereotype-incongruent and congruent trials using the following formulae:

$$Mid\ Phase\_Incongruent = MD\_Time\ incongruent\ trials - IT\_incongruent\ trials$$

$$Mid\ Phase\_Congruent = MD\_Time\ congruent\ trials - IT\_congruent\ trials$$

Then, the Mid Phase-Inc and the Mid Phase-Con were used to obtain the Mid Phase index for each participant according to the following formula:

$$Mid\ Phase\ index = Mid\ Phase\_Incongruent - Mid\ Phase\_Congruent$$

Similarly, the Late Phase index was obtained by computing the following steps:

$$Late\ Phase\_Incogruent = RT\_incongruent\ trials - MD\_time\ incongruent\ trials$$

$$Late\ Phase\_Congruent = RT\_congruent\ trials - MD\_time\ congruent\ trials$$

$$Late\ Phase\ index = Late\ Phase\_Incongruent - Late\ Phase\_Congruent$$

## Statistics

All the analyses were carried out in the R-studio (version: 1.1.463) environment employing ad-hoc built routines. To identify meaningful factors underlying the MT metrics, a PCA was conducted using all conditions of the IAT test (White-Positive, White-Negative, Black-Positive, Black-Negative). Once identified a recurrent factorial structure underlying all conditions, and verified structure consistency across conditions using Tucker's Phi [52], the most representative variables of each factor (hence called target observed variables) were identified.

PCAs with an Oblimin Rotation and Kaiser link (see Costello, Osborne [53] for a comprehensive review), was performed employing of the principal routine of the 'psych' R package [54] and the *pr.comp* routine of the 'stats' R package (R Core Team, [55]), while the Tucker's Phi coefficient was calculated using the *factor.conguence* function of the 'psych' R package [54].

In a second step, the target observed variables for each factor identified by PCA analysis were entered as dependent variables into a general linear mixed model (GLMM) ['lme4' R package [56], version: 1.1–21] with Type of attribute (Negative vs Positive) and Race (Blacks vs Whites) as fixed predictors, and subject ID as random intercept. The model with the best fit to the data was selected based on likelihood ratio tests and goodness of fit indexes [57]. If significant, the Type of attribute-by-Race interaction effect was further explored using pairwise comparisons while adopting an FDR correction. We then explored the relationship between the target observed variables with the RT-race-IAT scores. In particular, we computed the Δ value as the differences between target observed values collected in stereotypical-congruent vs stereotypical-incongruent trials.

The Δ values were included in a general linear model (GLM) to test their association with the RT-race-IAT D score (see S1 File for further details about the model's syntax). Then, we focused on distributional analyses of mouse-trajectories testing the bimodality of the distributions of the Z-score-transformed MD values collected across all stereotype-congruent and -incongruent trials [40] using the Hartigan's dip statistic (HDS; [40, 58]). The HDS method is based on the null hypothesis of unimodality. Therefore, if the test is significant, it is possible to reject the hypothesis that the distributional profile is unimodal. As a final step, we explored our three-phase model (see Fig 2) to better describe the temporal evolution of conflict processing during the MT-race-IAT. This approach gives us the chance, for the first time, to identify in which of the three phases the conflict between automatic and controlled responses can emerge. Accordingly, a one-sample Student's *t*-test was used to test whether the Early Phase index differed from zero. Then, we estimated a mixed model with Type of Phase (Early, Mid or Late) as fixed predictor and Subject ID as random intercept to explore the between-phase differences in the amount of decision conflict. Post-hoc results were corrected with Tukey's correction for multiple comparisons [56].

Finally, the above conflict processing phases data were then included in a GLM to test their association with the RT-race-IAT D score.

## Results

### Participants characteristics

Participants' handedness scores indicated a preference for the right hand for most participants (94 of 116 participants had scores higher than 60) and no strong preference for 22 participants (scores lesser than 60). Notably, all participants reported using a computer mouse (as indicated by the specific added item for computer mouse usage) exclusively with the right hand.

### RT-race-IAT D score

Overall our sample obtained a mean RT-race-IAT D score of 1.08 (standard error of mean = 0.04; one-sample Student's t-test (H0: μ = 0): $t_{(115)} = 26.43$, $p < .001$; for details see S1 File) indicating, on average, a pro-White implicit bias. It is worth noting that the D score collected in our sample is, on average, equivalent to that one reported in previous studies which used a similar RT-race-IAT structure [59–63].

### RTs and error rates comparison between RT- and MT-race-IAT

We expected that participants would complete the MT-race-IAT with a higher accuracy, but with longer reaction times as compared to the RT one, due to all the strategic motor corrections engaged during the temporal unfolding of the movements [17–20]. To test this hypothesis, a series of Wilcoxon's signed-sum test was used to compute differences in both accuracy and RTs between RT- and MT-race-IAT. Results showed a significant higher total accuracy ($W = 133.5$, $p < .001$, $r = −.575$) and longer RTs ($W = 966$, $p < .001$, $r = −.448$) in the MT-race-IAT compared to RT one (Table 4). This was the case in both stereotype-congruent [*Accuracy* ($W = 163.$, $p < .001$, $r = −.547$); *RTs* ($W = 1114$, $p < .001$, $r = −.412$)] and incongruent trials [*Accuracy* ($W = 521.5$, $p < .001$, $r = −.511$); *RTs* ($W = 960$, $p < .001$, $r = −.449$); Table 2].

### Principal component analysis of MT-race-IAT metrics

A Principal Component Analysis was performed on each condition of the MT-race-IAT (White-Positive, White-Negative, Black-Positive, Black-Negative), including all metrics collected. Seven metrics were included in every PCA. However, two of them (Entropy and Xpos-flips) were cut out due to their unsteadiness (Entropy) or the low contribution to the factorial structure in almost all conditions (Communalities of Xpos-flips: < 0.4).

**Table 2. RT- vs MT-race-IAT accuracy and reaction times comparison.**

| IAT Measures | RT-race-IAT | | | MT-race-IAT | | | Wilcoxon signed-rank test | | |
|---|---|---|---|---|---|---|---|---|---|
| | Median | Min | Max | Median | Min | Max | W | r | p |
| **Total Accuracy in %** | 0.93 | 0.79 | 0.99 | 0.97 | 0.82 | 1.00 | 133.5 | -0.575 | < .001** |
| **Accuracy Congruent Trials in %** | 0.96 | 0.85 | 1.00 | 1.00 | 0.91 | 1.00 | 163 | -0.547 | < .001** |
| **Accuracy Incongruent Trials in %** | 0.88 | 0.61 | 1.00 | 0.96 | 0.63 | 1.00 | 521.5 | -0.511 | < .001** |
| **Total Reaction Time in ms** | 848.61 | 592.63 | 1488.79 | 987.29 | 678.29 | 1711.43 | 966 | -0.448 | < .001** |
| **Reaction Time Congruent Trials in ms** | 845.93 | 589.63 | 1578.69 | 980.40 | 690.51 | 1708.67 | 960 | -0.449 | < .001** |
| **Reaction Time Incongruent Trials in ms** | 862.68 | 593.61 | 1561.65 | 993.73 | 664.08 | 1714.20 | 1114 | -0.412 | < .001** |

**Table 3. Eigenvalues (> 1), percentage of variance explained by each factor, and percentage of cumulative variance explained by the factorial solution with 2 factors in all the four conditions.**

| Condition | Factor | Eigenvalue | Variance Explained | Cumulative variance |
|---|---|---|---|---|
| *White-Positive* | 1 | 1.68 | 56.8% | |
| | 2 | 1.22 | 29.8% | 86.6% |
| *White-Negative* | 1 | 1.6 | 51.3% | |
| | 2 | 1.38 | 38.6% | 89.9% |
| *Black-Positive* | 1 | 1.60 | 51.3% | |
| | 2 | 1.38 | 38.3% | 89.6% |
| *Black-Negative* | 1 | 1.67 | 56.2% | |
| | 2 | 1.26 | 31.8% | 88% |

As reported in Table 3, two factors were isolated in all conditions, explaining a high percentage of variance (min. = 86.6%–max. 89.9%).

Based on the recurrent factorial structure extracted by the 4 PCAs (see Table 4), we attributed the first factor a temporal dimension, while the second component was more related to kinematic measures and thus represented a 'geometrical component'.

As supported by Tucker's Phi values, the factors isolated were consistent between-conditions (minimum value of congruence = -0.77, see Table 5).

Considering the above findings and the information extracted by the factorial structure in each condition, we have considered as 'target observed variables' (i.e., the most representative variables of each factor) those that showed the highest saturation scores in most of the conditions (see Table 4). Accordingly, in our further analyses, the MD variable has been considered for the geometrical component and the Acc-Max-Time variable for the temporal dimension.

**Table 4. Saturation in each factor, communalities, and uniqueness of all the metrics included in the PCAs run for each condition.**

| Condition | Metric | F1 | F2 | Communality | Uniqueness |
|---|---|---|---|---|---|
| *White-Positive* | RT | 0.83 | 0.03 | 0.67 | 0.32 |
| | AUC | 0.00 | **0.99** | 0.97 | 0.02 |
| | MD | 0.00 | **0.99** | 0.98 | 0.02 |
| | Vel-Max-Time | 0.90 | -0.03 | 0.82 | 0.18 |
| | Acc-Max-Time | **0.94** | 0.00 | 0.89 | 0.10 |
| *White-Negative* | RT | 0.87 | 0.10 | 0.77 | 0.23 |
| | AUC | 0.01 | **0.98** | 0.96 | 0.04 |
| | MD | -0.01 | **0.98** | 0.96 | 0.03 |
| | Vel-Max-Time | 0.94 | -0.04 | 0.88 | 0.12 |
| | Acc-Max-Time | **0.96** | -0.04 | 0.93 | 0.06 |
| *Black-Positive* | RT | 0.87 | -0.01 | 0.76 | 0.23 |
| | AUC | -0.08 | **0.98** | 0.96 | 0.03 |
| | MD | 0.08 | **0.98** | 0.96 | 0.03 |
| | Vel-Max-Time | 0.93 | 0.02 | 0.87 | 0.13 |
| | Acc-Max-Time | **0.96** | -0.01 | 0.92 | 0.08 |
| *Black-Negative* | RT | 0.87 | 0.15 | 0.71 | 0.29 |
| | AUC | -0.01 | 0.98 | 0.97 | 0.03 |
| | MD | 0.00 | **0.99** | 0.98 | 0.02 |
| | Vel-Max-Time | **0.91** | -0.01 | 0.83 | 0.16 |
| | Acc-Max-Time | **0.91** | -0.13 | 0.91 | 0.09 |

**Table 5. Tucker's Phi congruence value calculated for the two factors extracted on each couple of conditions.**

| Condition | F1 | F2 |
|---|---|---|
| *White-Positive / White-Negative* | 0.78 | -0.77 |
| *Black-Positive / Black-Negative* | 0.82 | -0.81 |
| *White-Positive / Black-Negative* | 1 | 0.99 |
| *White-Negative / Black-Positive* | 1 | 0.99 |
| *White-Positive / Black-Positive* | 0.81 | -0.81 |
| *White-Negative /Black-Negative* | 0.79 | -0.79 |

## MT metrics and race-IAT Congruency Effect

Basically, during the mouse movements, an IAT Congruency Effect emerges as a result of a higher attraction of the averaged-trajectories towards the incorrect response button in incongruent than congruent trials, mirroring the stronger conflict generated during the incongruent vs congruent categorisation task (see Fig 3-panel a). By the above evidence, we expected to obtain an evident race-IAT Congruency Effect using the target observed variables of geometrical and temporal dimensions (i.e., MD as the target observed variable for the geometrical factor and Acc-Max-Time for the temporal factor). Regarding the MD variable, GLMM results (see S1 File for further details about the model's syntax) showed a significant *main effect of type of attribute* ($\chi^2_{(1,464)} = 4.81$; $p = .028$) and a significant type of *attribute-by-race interaction effect* ($\chi^2_{(1,464)} = 119.54$; $p < .001$). We further explored this interaction, and we observed a typical IAT Congruency Effect characterised by higher MD values, thus higher attraction towards the incorrect response button, in associating negative attributes with Whites' faces than Blacks' ones ($\chi^2_{(1,464)} = -161.12$; $p_{FDR-corrected} < .001$) and lower MD values, in associating positive attributes Whites' faces than Blacks' ($\chi^2_{(1,464)} = 66.48$; $p_{FDR-corrected} < .001$; see Fig 3 panel-b). For what concerns the Acc-Max-Time variable, results showed a significant type of *attribute-by-race* interaction effect ($\chi^2_{(1,464)} = 632.05$; $p_{FDR-corrected} < .001$). In particular, we observed a typical IAT Congruency Effect characterised, even in this case, by higher Acc-Max-Time values in pairing negative attributes with Whites' faces than Blacks' ones ($\chi^2_{(1,464)} = -129.24$; $p_{FDR-corrected} < .001$) and lower Acc-Max-Time values in pairing positive attributes Whites' faces than Blacks' ($\chi^2_{(1,464)} = 137.40$; $p_{FDR-corrected} < .001$; see Fig 3 panel-b).

## MT metrics and RT-race-IAT D score

Considering the above findings showing an evident race-IAT Congruency Effect using MD and Acc-Max-Time variables, we tested whether both metrics may be used to evaluate the individual implicit racial bias as much as the classical D scores. We explored the relationships between the above metrics with the RT-race-IAT scores to test this hypothesis. As reported in the Method section, we computed the Δ values for the MD and the Acc-Max-Time according to the following formulae:

$$\Delta - MD = MD\_incongruent\ trials - MD\_congruent\ trials$$

$$\Delta - Acc - Max - Time = Acc\_Max\_Time\_incongruent\ trials - Acc\_Max\_Time\_congruent\ trials$$

The results of the GLMs showed that RT-race-IAT D score is a significant predictor of both Δ–MD ($b = 35.73$, $t_{(112)} = 3.42$, $p < .001$; see Fig 3 panel-c;) and Δ-Acc-Max-Time ($b = 44.97$, $t_{(112)} = 3.08$, $p = .002$; see Fig 3 panel-c; for details see S1 File).

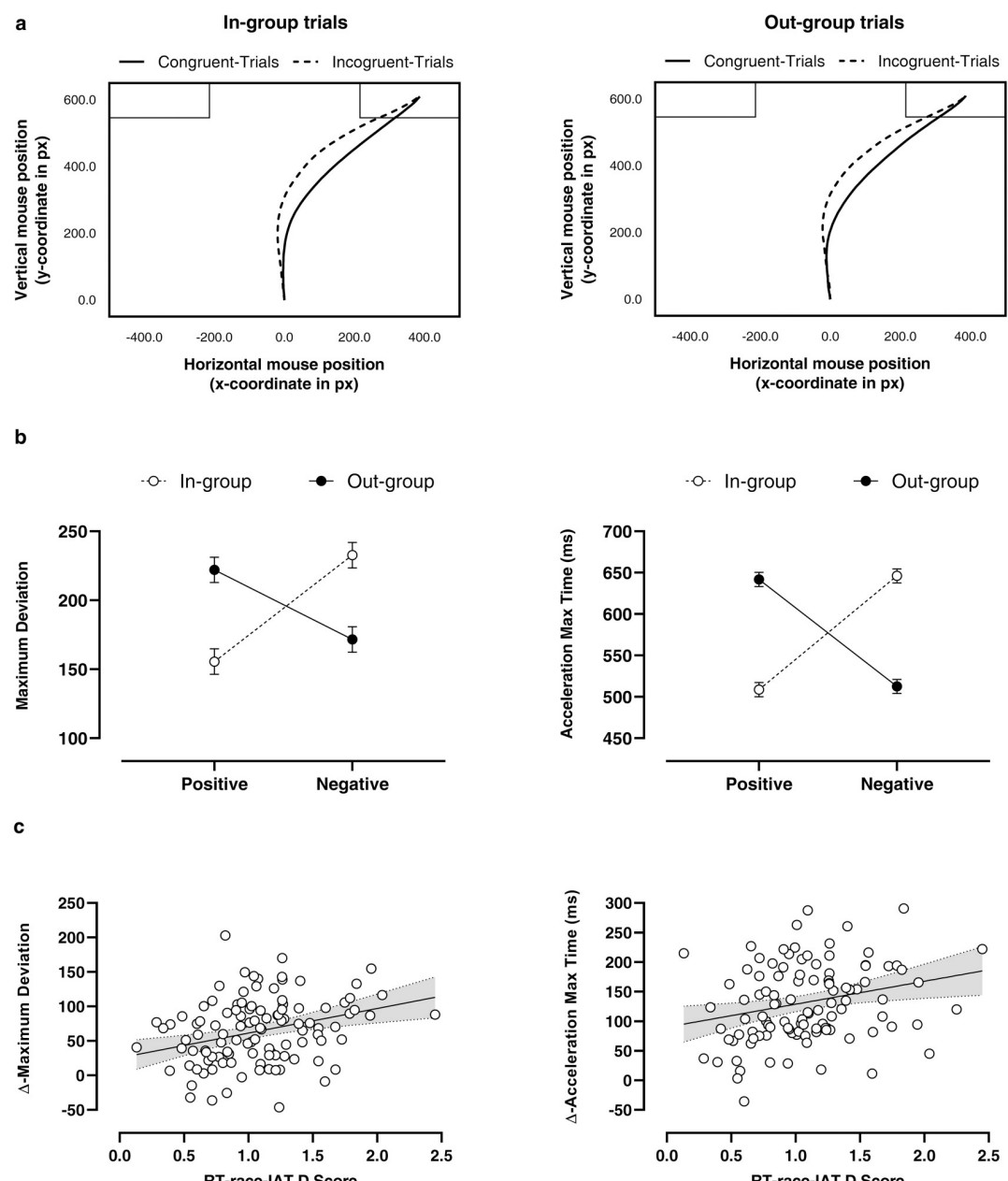

**Fig 3.** Panel (**a**), average real-time mouse trajectories for congruent and incongruent in-group trials (left panel) and congruent and incongruent out-group trials (right panel) in the MT-race-IAT task. Panel (**b**), MT-metrics differences collected during the experimental task. The line-plot graphs represent the Maximum Deviation (left panel) and Acceleration-Max-Time (right panel) data as a function of attribute type and actor's race. Values are expressed as mean ± SEM. Panel (**c**), linear regressions between RT-race-IAT D Score vs Δ-Maximum Deviation (left panel) and Δ-Acceleration-Max-Time (right panel); the solid line indicates the best linear model, and the dashed lines indicate the 95% confidence intervals.

## Distributional analysis of MT trajectories

As previously reported, we tested the bimodality of the distributions of the MD values collected across all congruent and -incongruent trials [40] by computing Hartigan's dip statistics (HDS; [40, 58]) on the Z-score-transformed MD distributions (Z-MD). The results showed no

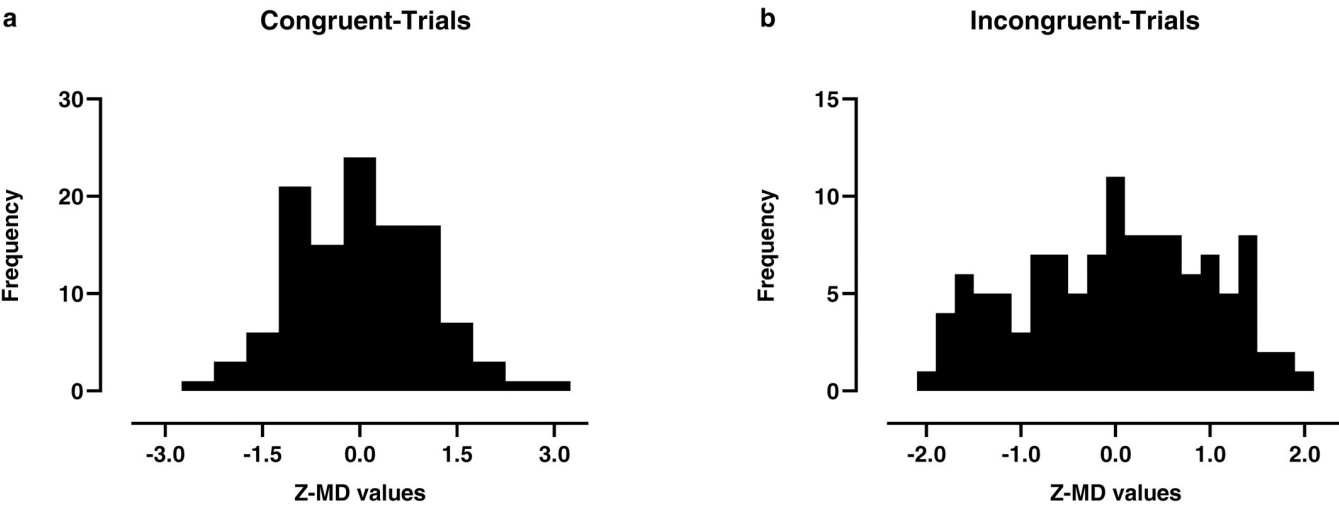

**Fig 4.** Z-score-transformed MD frequency distributions in stereotype-congruent (**a**) and incongruent (**b**) trials.

evidence for bimodality of Z-MD in either congruent ($d$ = .026, $p$ = .87) or incongruent trials ($d$ = .025, $p$ = .89; see Fig 4), suggesting that the decision process and the consequently raised conflict appear to unfold in a smooth manner reflecting a single dynamical cognitive processing.

## Conflict's temporal unfolding phases

Considering the above evidence, we want to temporally dissect this continuous monomodal dynamical competition process using the three-phase model described above (Fig 2). Our analysis revealed the existence of a clear race Congruency Effect already during the early phase (standard error of mean = 5.03; one-sample Student's t-test (H0: μ = 0): $t_{(115)}$ = 6.66, $p < .001$). In addition, the GLMM analysis showed a significant main effect of phase ($\chi^2_{(2,345)}$ = 83.14; $p_{FDR\text{-}corrected} < .001$; see S1 File for further details about the model's syntax) with the highest race Congruency Effect observed in the Mid Phase (Early vs. Mid $p_{Tukey\text{-}corrected} < .001$, Early vs. Late $p_{Tukey\text{-}corrected}$ = .451, Mid vs Late $p_{Tukey\text{-}corrected} < .001$; see Fig 5).

## Conflict's temporal unfolding phases and RT-race-IAT D score

Results showed that RT-race-IAT D score is a significant predictor of the Early ($b$ = 30.29, $t_{(112)}$ = 2.77, $p$ = .006; see Fig 6A) and the Mid Phase ($b$ = 27.83, $t_{(112)}$ = 2.30, $p$ = .023; see Fig 6B), but not of the Late Phase ($b$ = 10.36, $t_{(112)}$ = 1.00, $p$ = .318; see Fig 6C, for more details see S1 File). This evidence suggests that the classical RT-race-IAT D score merges the first two different phases of conflict detected by our MT-race-IAT.

## Discussion

The purpose of the present study was to incorporate MT-measures in a classical race-IAT paradigm [42] to go beyond RTs and error rates, gathering an in-depth characterisation of the individual implicit bias and following the temporal unfolding of the decision conflict behind the automatic vs controlled categorisation responses in a race-IAT [22, 25–29, 39].

Differently from previous experimental works which have adopted the MT approach in an IAT structure [31, 32], we decided to use a classical within-subject design to have the chance to directly compare participants' performance gathered in a standard keyboard-response race-

# Cumulative Congruency Effect

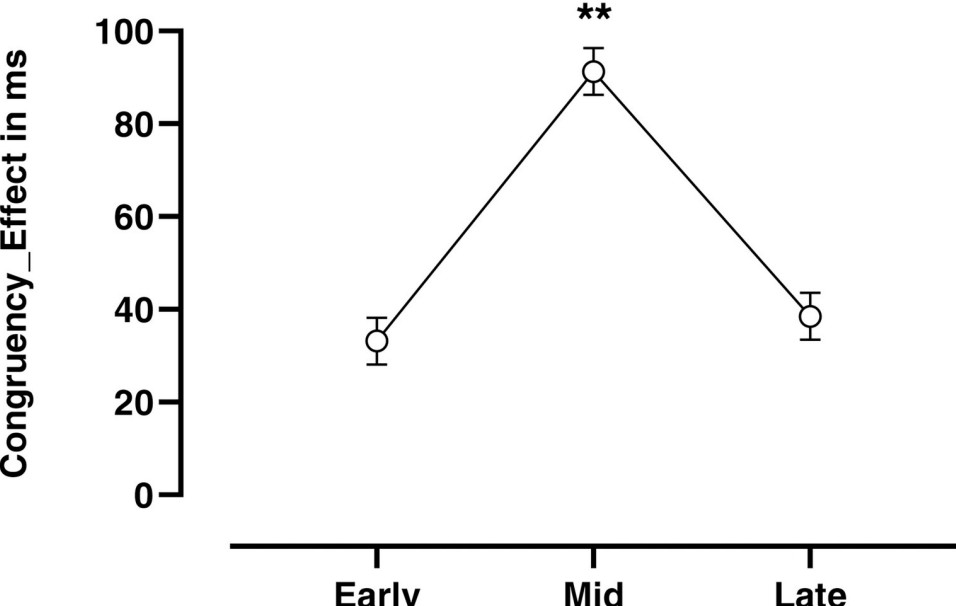

**Fig 5. Conflict's temporal unfolding phases.** The line-plot graphs represent the Early, Mid and Late phases, according to which the temporal unfolding of the decision towards the target choice in the MT-race-IAT was dissected. Values are expressed as mean ± SEM; $^{**}p < .001$ (corrected for Tukey's correction for multiple comparisons).

IAT (RT-race-IAT) with the MT one. Accordingly, from a behavioural point of view, our participants obtained a mean RT-race-IAT D score of 1.08, indicating, on average, a pro-White implicit bias to the extent that was comparable with previous reports which used a similar RT-race-IAT structure [59–63]. In our model, the direct comparison between the accuracy and reaction times (measured in the overall task and critical blocks) collected in the RT- vs the MT-race-IAT task revealed a significantly higher accuracy and longer RTs in the MT-race-

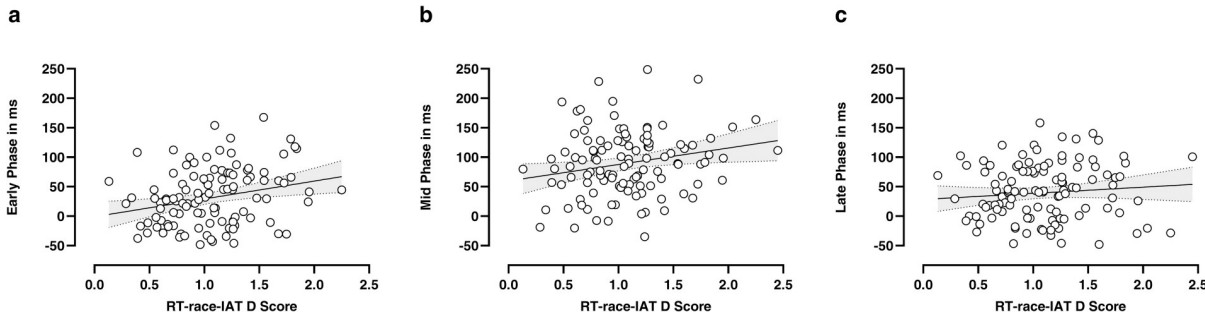

**Fig 6.** Linear regressions between RT-race-IAT D Score vs Early (**a**), Mid (**b**) and Late (**c**) conflict's temporal unfolding phases. The solid line indicates the best linear model, and the dashed lines indicate the 95% confidence intervals.

IAT (see Table 2). This evidence is consistent with recent findings [15, 22, 25, 26, 64] and emphasises the impact of the strategic motor corrections engaged during the temporal unfolding of the hand movements to reach the selected category [17–20].

Among all the MT metrics, our PCA data identified two target variables, MD and Acc-Max-Time, as the most representative of geometrical and temporal dimensions, respectively (see Tables 3–5). These were considered the more suitable empirical measures to estimate the race Congruency Effect in this experiment. Results showed a typical race Congruency Effect characterised by, on average, higher MD and longer Acc-Max-Time values in associating negative attributes with Whites' faces than Blacks' ones, mirrored by lower MD and Acc-Max-Time values in associating positive attributes to Whites' faces than Blacks' (see Fig 3 panel-b). These findings are consistent with previous evidence showing a Congruency Effect using the MT approach in other IAT structures [31, 32]. As a novel contribution, taking advantage of our within-subject design, we could explore the possible correlation between the RT-race-IAT D score with MD and Acc-Max-Time values differences collected in incongruent vs congruent trials. Our data showed that RT-race-IAT D score was positively associated with both MD and Acc-Max-Time values (see Fig 3 panel-c), suggesting that the MT approach can characterise and investigate, yet in a more detailed fashion (as will be described below), implicit bias as classical RT-race-IAT D score [42].

In the last decade, the MT approach has proven to be particularly powerful in shedding light on the dynamics behind 'decision conflict' resolution in a categorisation task [15, 21, 22]. Several neuropsychological studies considered hand movement as a proxy to detect and quantify the intertwined automatic and controlled strategic motor corrections needed to update actions based on cognitive evaluations that rapidly occur during the decision process [17–20, 27–29]. Traditionally, the MT approach has always been used to rule out which class of cognitive frameworks has been engaged in guidance the temporal evolution of judgements and decisions [26]. The dual-system or stage-based perspective is the most widely endorsed theoretical framework used to make predictions about the temporal sequence of a decision process towards a target choice [33, 34]. This framework implies that a decision unfolds via a two-system sequence whereby a faster, automatic driven representation (system 1) guide the early path of hand movement towards the target choice, and, after a time lag, a slower cognitive system (system 2) can exert a controlled response to correct this initial response [15, 22, 26–29, 33, 34, 65]. Besides the dual-system perspective, another class of theoretical model refers to a dynamical competition (Dynamical Framework) of different controlled and automatic representations that synchronously interact in guiding the path of hand movement towards the target choice [27–29, 35, 36, 66]. Notably, the distributional analyses of the MT metrics allow detecting whether the trajectories follow an abrupt profile, characterised by a sharp mid-flight correction mirroring the competition between two systems during the evolution of the movement (Dual-system framework; [33, 34]), or relatively smooth profile mirroring a dynamical competition of different controlled and automatic processes that simultaneously guide the movement (Dynamical framework; [35, 36, 66]). In line with Yu, Wang [31], the distributional analyses on our MT-race-IAT showed that in both congruent and incongruent trials, mouse trajectories displayed smooth fashion profile indicating that, overall, the decision process and the consequently raised cognitive conflict appear to unfold guided by dynamical cognitive processing (see Fig 4).

This latter finding is also in line with the 'conflict monitoring hypothesis' [27–29], strengthening the need to better characterize this continuous monomodal dynamical competition by studying its unfolding and identifying behavioural markers of the imbalance between automatic and controlled processes over time. Accordingly, our novel phase-based approach allowed us to temporally dissect this continuous decision conflict by combining different

temporal MT-metrics (IT, MD-Time and RT). From a mechanistic point of view, the evolution of the race Congruency Effect in an MT-race-IAT trial can be divided into the following temporal intervals. On each trial, participants clicked a 'start' button appearing at the bottom centre of the screen, and the target stimulus appeared in its place; from that moment, an automatic default response is predominantly activated in line with the '*direct pathway*' suggested by Botvinick, Braver [27] and Erb, Moher [29]. The latency between stimulus presentation and actual mouse movement was considered the first temporal interval, namely the Early Phase. Once the movement is started, the dynamical competition between automatic and controlled responses progressively increases, reaching its apex at the point of maximum attraction towards the incorrect response [15, 18–20, 27, 29]. The latency between the beginning of mouse movement to the competition's apex corresponded to the Mid Phase. At this stage, in the incongruent trials, the controlled motor corrections progressively allow participants to reach the target, thus concluding the trial. This latter latency in reaching the target response from the highest attraction towards the incorrect response accounted for the third and last interval, namely the Late Phase [26] (a schematic illustration of the above phases during the temporal unfolding of an MT-race-IAT trial is displayed in Fig 2). Our data showed that a detectable race Congruency Effect is already present during the Early Phase of stimulus processing (see Fig 5). This evidence is particularly intriguing since it shows that, in accordance with the associative network model introduced by Greenwald, Banaji [5], during the incongruent trials the weak association between social concepts and attribute concepts impacts the stimulus processing kinetics by inducing a significant delay and thus cognitive conflict prior an actual mouse movement [29]. This suggests that a 'decision conflict' is detectable even in the early phase of incongruent trials. Besides, we show that the initial amount of conflict (Early Phase) significantly increased across the Mid Phase, going back to the Early Phase level during the Late Phase (see Fig 5). Here we show i) that the Mid Phase displays the highest level of race Congruency Effect as compared with the other two phases, thus confirming that in this interval, the decision conflict between automatic and controlled responses reached its maximum degree; ii) the existence of a significant race Congruency Effect also during the Late Phase, namely after the maximum deviation point which was traditionally considered to reflect the moment of final conflict resolution [15, 25], suggests that the decision conflict between automatic and controlled responses is still detectable, even if smaller, once the correct response is selected. Most likely, this latter aspect could not be revealed using a traditional RT-race-IAT. Indeed, RT-race-IAT D scores were positively associated with both Early- and Mid-Phase values but not with the Late Phase ones (see Fig 6). This last evidence shows that by means of the higher temporal resolution of the above phase-based approach, a finer-grained characterisation of individual racial bias can be achieved, suggesting that in the classical RT-race-IAT D score are collapsed two different phases of conflict processing in a race-IAT, and it is not possible to gauge the late phase of that processing. The analysis of the above temporal phases has clearly pointed out that the developed phase-based approach can be considered a powerful novel tool to insight into conflict processing unfolding during a race-IAT task and, most notably, to reach a fine-grained characterisation of individual racial bias.

The present study is not free from limitations. In this regard, although we draw our conclusion recruiting a relatively large sample size with a reliable homogeneity both for age and ethnicity (see S1 Table in S1 File), to generalise our findings, future studies should consider testing our phase-based approach with larger and more representative samples. In addition, since we decided to use a classical within-subject design to directly compare participants' performance gathered in the classical RT-race-IAT with the MT one, we did not counterbalance the temporal order of the *congruent* and *incongruent* combined tasks. It is worth noting that an *order effect* caused by *cognitive inertia* may affect the IAT effect, especially in a between-subject

design [67, 68]. Therefore, future studies that aim to employ our phase-based approach in a between-subject paradigm should consider counterbalancing the order of the experimental blocks, thus mitigating the potential *order effect* (for details concerning the best research practices for using the IAT see [67]).

In conclusion, in the present study, we show the first evidence that the MT approach can characterise and investigate, yet in a more detailed fashion, the same racial bias construct as traditionally studied employing the classical RT-race-IAT D score [42], by using a phase-based approach that can be considered a compelling novel tool to insight into conflict processing unfolding during a categorisation task with a promising transferable value in different cognitive and neuropsychological fields.

## Supporting information

**S1 File. Supporting information contains all the supporting tables and figures.** (DOCX)

## Acknowledgments

We want to show our gratitude to Umberto Manganiello, Software Engineer, Bank of Italy, who greatly assisted the research providing insight and expertise in developing the loop-operations implemented in the ad-hoc R routines used to analyse the mouse-tracking data.

## Author Contributions

**Conceptualization:** Michael Di Palma, Andrea Minelli, Manuela Berlingeri.

**Data curation:** Michael Di Palma, Desiré Carioti.

**Formal analysis:** Michael Di Palma, Desiré Carioti.

**Investigation:** Michael Di Palma, Elisa Arcangeli, Andrea Minelli.

**Methodology:** Michael Di Palma, Desiré Carioti, Andrea Minelli, Manuela Berlingeri.

**Project administration:** Michael Di Palma, Andrea Minelli, Manuela Berlingeri.

**Supervision:** Andrea Minelli, Manuela Berlingeri.

**Validation:** Michael Di Palma, Cristina Rosazza, Patrizia Ambrogini, Riccardo Cuppini, Andrea Minelli.

**Writing – original draft:** Michael Di Palma, Desiré Carioti, Andrea Minelli, Manuela Berlingeri.

**Writing – review & editing:** Michael Di Palma, Andrea Minelli, Manuela Berlingeri.

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
