## [Decision Letter · Decision Letter 0]

5 Jan 2022

PONE-D-21-32162The biased hand. Mouse-Tracking metrics to examine the underlying conflict processing in a race-implicit association testPLOS ONE

Dear Dr. Berlingeri,

Thank you for submitting your manuscript to PLOS ONE. After careful consideration, we feel that it has merit but does not fully meet PLOS ONE’s publication criteria as it currently stands. Therefore, we invite you to submit a revised version of the manuscript that addresses the points raised during the review process.

Both reviewers raise a number of highly relevant theoretical points that need to be addressed. Particularly when it comes to the concepts of bottom-up and top-down, conscious/ unconscious and implicit/explicit the manuscript needs to be clearer and more precise in what is actually meant by these concepts, and how they relate to the design and results that are reported here. Rigorously revising your manuscript so that it has a stronger and specifically unambiguous/ clear theoretical embedding will increase the readability and the impact of this paper. Reviewer 1 also raises a relevant point with regards to the design: if the blocks are indeed not counterbalanced, then this has important consequences for how the results can be interpreted (i.e. whether they are potentially just sequence effects). If there was no counterbalancing, I would urge you to run a follow-up experiment that does counterbalance the sequence of the blocks over participants, to ensure that your current results are not confounded. ==============================

We look forward to receiving your revised manuscript.

Kind regards,

Marte Otten, Ph.D.

Academic Editor

PLOS ONE

Journal Requirements:

3. Please note that according to our submission guidelines (http://journals.plos.org/plosone/s/submission-guidelines), outmoded terms and potentially stigmatizing labels should be changed to more current, acceptable terminology. For example: “Caucasian” should be changed to “white” or “of [Western] European descent” (as appropriate).

Reviewers' comments:

Reviewer's Responses to Questions

**Comments to the Author**

1. Is the manuscript technically sound, and do the data support the conclusions?

Reviewer #1: Partly

Reviewer #2: Partly

2. Has the statistical analysis been performed appropriately and rigorously? 

Reviewer #1: N/A

Reviewer #2: N/A

3. Have the authors made all data underlying the findings in their manuscript fully available?

Reviewer #1: Yes

Reviewer #2: No

4. Is the manuscript presented in an intelligible fashion and written in standard English?

Reviewer #1: Yes

Reviewer #2: Yes

5. Review Comments to the Author

Reviewer #1: This study presents results of original research investigating conflict processing in a race IAT (Caucasian/African; positive/negative) among Caucasian native Italian speakers. Participants completed both a classical, reaction-time based race IAT and a mouse-tracking (MT) version. The manuscript is overall well-written, although I would like the authors to perform additional proofreading as several typos and minor errors are left in the manuscript. 120 participants took part in the study (a few excluded) and the authors seem to suggest that sample size was determined based on previous research using a MT version of the IAT. Although this rule of thumb may be sometimes warranted, I would have expected a more stringent a priori power analysis. This being said, I have several comments, mainly regarding (lack of) theoretical justification and some related to methods and analyses.

1/p.5: Given the theoretical background mobilized by the authors, I was wondering whether it is appropriate to state that "motor corrections [are] needed to connect cognition to action" -> this may (artificially) distinguish action from cognition, while we may consider the two are so intrinsically connected that this distinction is difficult to defend. What do the authors think?

2/p.5: Please precisely define your definitions of bottom-up and top-down processes (including “processes”) - these terms are sufficiently equivocal to raise debates (with some scholars questioning their relevance altogether).

3/p.5: Please provide a much more rigorous and convincing rationale for you 3-phase choice: this is an important feature of your work that is currently ill-justified -> why are 3 phases needed - compared to 2, 4 or an infinity (at the theoretical level, as this is not a trivial choice for MT)? Indeed, beneath the surface (appealing simplicity), this 3-phase categorization has the potential to jeopardize the very advantages of mouse-tracking (providing continuous signals).

4/p.5: Stating that the "early phase (...) [may comprise] purely implicit, bottom-up mechanisms" is currently inappropriate. Please refer to recent papers discussing the problems associated with clearly defining purely "implicit" processes. Second, by no means should implicit be equated with bottom-up processes (as I understand them but, again, these need to be defined). Third, past research relies on mouse-tracking to question the implicit/explicit distinction and some of the study’s results go in the same direction. So, to avoid incorrectness/overstatement, please stay close in your wording to what this early phase may plausibly reflect.

5/p.5: As the terms conscious and unconscious processes are often debated, either they need to be defined or should be replaced (I am for instance doubting that JB Freeman uses the conscious/unconscious terminology, except maybe in some of his early work). Later on, the authors use "unconscious, purely implicit, bottom up processes". I urge the authors to be both more cautious and precise in their wording. They are not equivalent according to current literature and are equivocal without further clarification.

6/p.6: I would have expected a better integration of this bimodality/unimodality aspects with the authors' phases model. Could the authors please provide some insights?

p.9: It seems like the authors did not counterbalance critical congruent and incongruent blocks. If not, this is a serious issue, both experimentally (problem in design) and statistically (incorrect estimation of effect sizes - here, artificially increasing effect sizes). Could the authors please clarify and provide appropriate justification?

7/p.12: Why only subject ID, and as random intercept only (what about slopes?) and what about stimuli? Please provide a strong justification of your selected model and, if changes in random effects impact your main findings, this should be acknowledged.

Reviewer #2: The authors adapted the race IAT to the Mouse-Tracking setting and explored the relationship between the race IAT and metrics used in Mouse-Tracking. They found that some metrics from adapted race IAT in Mouse-Tracking also showed the congruency effects from the regular IAT. They also used a novel way to segment the Mouse-Tracking metrics by its temporal unfolding and examined their relationship with the IAT score. I appreciate the authors attempts to compare two methods that are both able to capture the more automatic and uncontrollable processes yet are usually studied under different frameworks. However, I have several major questions about the work.

First, regarding the utility of Mouse Tracking for studying race biases, in addition to the citations included in the paper, Melnikoff et al., 2020 showed Mouse-Tracking are useful in studying racial biases. It predicted behavioral racial biases. Although it did not compare IAT test and Mouse-Tracking directly, it makes me wonder what is new about this manuscript. There are a lot of descriptive results in the manuscript, but it’s unclear to me what is the main question this research is trying to solve. This is partly due to my questions (see below) about the two main points the authors are trying to demonstrate in the paper.

Second, in the intro and discussion, the authors mentioned multiple times about bottom-up and top-down processes and seem to suggest that Mouse-Tracking results can help to adjudicate the different processes. This is also one of the main goals of the paper. Maybe I misunderstand the authors intention, but I’m a bit confused about this theoretical point. What are top-down and bottom-up conflict, respectively in the Mouse-Tracking context? And how and what question is the three phases model can help to solve?

Third, could the authors help me to understand the reasons behind dissecting by temporal mark? Also, after the factor analysis, the authors picked one metric from each factor to follow up. However, the other unpicked metrics show similar quality to the picked ones. What would the results be if the other metrics from the two factors are used? This makes me question whether the second goal of this paper, identify metrics from Mouse-Tracking that is comparable to the IAT metrics, is achieved properly.

Fourth, one thing Mouse-Tracking is inherently different from IAT is that people are explicitly evaluate the targets in Mouse-Tracking (more like self-report). In a standard Mouse-Tracking task, people can take their time to reach a final decision and this is part of the RT data. However, IAT sometimes pose a time limit on responses or take care of the longer time when calculating D score. Does the authors take some steps in the work to alter the design of the Mouse-Tracking task to the IAT test?

6. PLOS authors have the option to publish the peer review history of their article (what does this mean?). If published, this will include your full peer review and any attached files.

Reviewer #1: No

Reviewer #2: No

---

## [Author Response · Author response to Decision Letter 0]

23 Feb 2022

In the file ‘Di Palma et al - Response to Reviewers’, we report point-by-point responses to all Reviewers’ concerns and suggestions.

---

## [Editor Report · Decision Letter 1]

24 May 2022

PONE-D-21-32162R1The biased hand. Mouse-Tracking metrics to examine the underlying conflict processing in a race-implicit association testPLOS ONE

Dear Dr. Berlingeri,

Thank you for submitting your manuscript to PLOS ONE. After careful consideration, we feel that it has merit but does not fully meet PLOS ONE’s publication criteria as it currently stands. Therefore, we invite you to submit a revised version of the manuscript that addresses the points raised during the review process.

Thank you for the revised manuscript. Unfortunately, the original reviewers were not available anymore to evaluate whether your revisions met their requirements. I have reviewed the revised manuscript myself: with the implemented changes it meets the publication requirements with regards to the manuscript. However, without access to the original raw data (so not the results that are in the manuscript and supplements) I cannot accept this manuscript. Please make sure that the original datafiles on which your analyses are based are available and accessible (obviously after they are anonymised). If that is taken care off, I can accept the manuscript as it is.

We look forward to receiving your revised manuscript.

Kind regards,

Marte Otten, Ph.D.

Academic Editor

PLOS ONE

Journal Requirements:

Additional Editor Comments (if provided):

Thank you for the revised manuscript. Unfortunately, the original reviewers were not available anymore to evaluate whether your revisions met their requirements. I have reviewed the revised manuscript myself: with the implemented changes it meets the publication requirements with regards to the manuscript. However, without access to the original raw data (so not the results that are in the manuscript and supplements) I cannot accept this manuscript. Please make sure that the original datafiles on which your analyses are based are available and accessible (obviously after they are anonymised). If that is taken care off, I can accept the manuscript as it is.
---

## [Author Response · Author response to Decision Letter 1]

6 Jul 2022

In the file ‘Di Palma et al - Response to Additional Editor Comments’, we report the response to the additional Editor's comments.

---

## [Editor Report · Decision Letter 2]

7 Jul 2022

The biased hand. Mouse-Tracking metrics to examine the conflict processing in a race-implicit association test

PONE-D-21-32162R2

Dear Dr. Berlingeri,

We’re pleased to inform you that your manuscript has been judged scientifically suitable for publication and will be formally accepted for publication once it meets all outstanding technical requirements.

Kind regards,

Marte Otten, Ph.D.

Academic Editor

PLOS ONE
---

## [Editor Report · Acceptance letter]

18 Jul 2022

PONE-D-21-32162R2 

The biased hand. Mouse-Tracking metrics to examine the conflict processing in a race-implicit association test 

Dear Dr. Berlingeri:

I'm pleased to inform you that your manuscript has been deemed suitable for publication in PLOS ONE. Congratulations! Your manuscript is now with our production department. 

Kind regards, 

on behalf of

Dr. Marte Otten 

Academic Editor

PLOS ONE